# *Eucalyptus cinerea* and *E. nicholii* by-Products as Source of Bioactive Compounds for Agricultural Applications

**DOI:** 10.3390/plants11202777

**Published:** 2022-10-20

**Authors:** Paola Malaspina, Marina Papaianni, Marta Ranesi, Flavio Polito, Cristina Danna, Pierluca Aicardi, Laura Cornara, Sheridan L. Woo, Vincenzo De Feo

**Affiliations:** 1Department of Earth, Environment and Life Sciences, University of Genova, Corso Europa 26, 16132 Genova, Italy; 2Department of Agricultural Sciences, University of Naples Federico II, Via Università 133, 80055 Portici, Italy; 3Department of Pharmacy, University of Salerno, Via Giovanni Paolo II, 132, 84084 Fisciano, Italy; 4Coldiretti Savona, Piazza Giulio II 4-7, 17100 Savona, Italy; 5Department of Pharmacy, University of Naples Federico II, Via Domenico Montesano 49, 80131 Naples, Italy

**Keywords:** waste reuse, plant metabolites, essential oils, micromorphology, alternative pesticides, phytotoxicity

## Abstract

The cultivation of different species of *Eucalyptus* has recently expanded in Liguria (Italy) due to the growing demand of the North European floricultural market. Eucalyptus tree branches are cut and selected for their quality, resulting in large amounts of waste biomass to be disposed of. The aim of our study was to evaluate the phytotoxic and antimicrobial activities of essential oils (EOs) from pruning wastes of *E. cinerea* (EC) and *E. nicholii* (EN), for potential applications in agriculture. Phytochemical analyses showed eucalyptol (1,8-cineole) as the major component in both EOs, but the EO yield of EN was higher than that of EC, in agreement with a significantly higher oil gland density on EN leaves. EOs from both species showed phytotoxicity on both weeds tested, but no significant inhibition on horticultural crop seed germination, except for *Raphanus sativus*. The EO from EC showed the strongest antibacterial activity, while the EO from EN showed the strongest antifungal activity. Concluding, EOs from *Eucalyptus* pruning may be used as possible alternatives to synthetic herbicides and pesticides, acting as antimicrobial and antifungal agents, thus representing a safe strategy for crop management programs.

## 1. Introduction

In Liguria, the Western Riviera is also known as the Flower Riviera due to its dedication to floriculture since the early nineteenth century [1]. In the last few decades, the cultivation of *Eucalyptus* has expanded and developed in this region, and presently the export of ‘eucalyptus cut foliage’ is growing on the Northern European floriculture market, where fresh and dried branches are used in floral arrangements. The fronds selected for this ornamental purpose must have uniform visual characteristics regarding the length of the branches, shape, color, and waxiness of the leaves, and therefore during harvesting, the cut branches are carefully selected for their quality and uniformity.

As a consequence of this culling operation, a large amount of waste biomass is produced (more than 4 tons per year) in the Province of Savona, representing an economic and social issue. Currently, pruning biomass is eliminated by burning or burying [2], resulting in the loss of bioactive compounds contained in the vegetative by-products that potentially may have a high biological value. Indeed, these often-overlooked matrices can be a rich source of valuable secondary metabolites, such as phenolic compounds, waxes, and essential oils, which can be used for various applications in food, medicinal, or agricultural sectors. Therefore, the management of *Eucalyptus* by-products can be of primary importance to reduce the volume of waste accumulated in landfills and to develop strategies for the recovery and enhancement of value-added products [2,3,4].

Recent research indicated that *Eucalyptus* essential oils (EOs) possess strong antimicrobial activity and therefore have wide potential applications in the medical, food, and chemical industries [5,6,7]. Additionally, *Eucalyptus* EOs are also known for their herbicidal, insecticidal, and acaricidal activities [8,9].

In a previous study, we provided useful information on the phytotoxic properties of the EOs obtained from remnants of *E. gunnii* Hook. f. and *E. pulverulenta* Sims cv ‘Baby blue’ used as natural herbicides [2]. The present work will focus on the by-products of two different *Eucalyptus* species, *E. cinerea* F. Muell. ex Benth. (EC) and *E. nicholii* Maiden and Blakely (EN), widely cultivated in western Liguria for ornamental purposes. Macro- and micromorphological features of the leaves, chemical composition, and biological properties of the EOs, including phytotoxic, antibacterial, and antifungal activities, will be investigated in order to explore the possibility of recovering bioactive molecules for agricultural applications.

## 2. Results

### 2.1. Micromorphological Characterization

The leaves of *E. cinerea* (EC) and *E. nicholii* (EN) are both glabrous and leathery, but they show different shapes and variable colors depending on the abundance and arrangement of waxes. EC leaves are opposite, sessile, oval/rounded, blue-gray, and densely reticulated. EN leaves are alternate, pedunculated, elongated, and dark green. Through stereomicroscopy (Figure 1A), EC showed a layer of waxes covering the oil glands and making them less visible. Differently, in EN, many oil glands containing EOs emerge on the leaf surface (Figure 1B). Leaf epidermal peelings analyzed via light microscopy (LM) (Figure 1C,D) showed a significantly lower oil gland mean density in EC with respect to EN. In addition, in each species, significant differences in oil gland density were also found between the adaxial and abaxial surfaces of the leaf (Table 1). In leaf transversal sections, observed with both LM and SEM, the cavities of oil glands appear spherical to ellipsoidal in shape. They were scattered in the mesophyll, and especially located in the subepidermal region on both sides of the leaves (Figure 2). In both the upper and lower epidermis, two overlaying cells are located exactly over the secretory cavities, differing in shape, size, and color from the other epidermal cells (Figure 1C,D, arrows; Figure 3C,D, arrows).

The midrib, in the transversal section, is slightly convex on both sides and consists of one or two large bicollateral vascular bundles in an open arc, with dorsal trace types more or less evident (Figure 3). The leaves of both species show a uniseriate epidermis with a smooth and thick cuticle and well-visible papillae (Figure 3A–D). As frequently occurring in *Eucalyptus* species, leaves are amphistomatic with both anomocytic and actinocytic stomata (Figure 3C,D).

Crust waxes were found in both species: tubular waxes in EC and crystalloid waxes (rosettes) in EN (Figure 3E,F).

### 2.2. Chemical Composition of Essential Oils

The hydrodistillation of the leaves of EC and EN furnished pale yellow EOs with a yield of 2.56% and 3.67%, respectively, on a fresh weight basis (Table 1). The yield is an important parameter to be considered because the possibility to extract EOs from waste material to reuse the by-products in a perspective of the circular economy is promising. These EO yield data are in agreement with the abundance of the oil glands shown through micromorphological analysis. The putative composition identity of the EOs with retention indices and area percentages for each component from the MS runs are reported in Table 2. The compounds are listed according to their elution order on an HP5-MS column.

Altogether, 31 components were identified, 28 for the essential oil of EC, accounting for 97.6% of the total EO, and 22 for EN, accounting for 97.7% of the total EO.

In both EOs, eucalyptol was the main component, accounting for 67.7% in EC and for 79.5% in EN. In both EOs, oxygenated monoterpenes predominate, totaling 78.0% and 85.0%, respectively, for EC and EN.

In the EO from EC, the other components comprising a percentage greater than 1% are α-pinene (7.3%), and terpinene derivatives: α-terpinene (3.3%), γ-terpinene (1.2%), terpinolene (3.5%), α-terpineol (3.9%) and α-terpinyl acetate (5.2%).

The essential oil of EN is very similar to that of EC apart from the absence of α-terpinyl acetate. Additionally, in this EO, the main constituents were α-pinene (3.7%), α-terpinene (3.3%), terpinolene (2.5%), and α-terpineol (3.9%).

In both EOs, the sequiterpene fraction is very poor (2.1% and 0.7%, respectively, for EC and EN).

### 2.3. Phytotoxic Activity

Table 3 and Table 4 report the activity of the EOs from EC and EN on the germination of diverse seeds. The EOs showed variable phytotoxicity on the seeds depending on the plant species tested and the concentration used. In general, these EOs were phytotoxic to both weeds tested, negatively affecting the seed germination and radicle elongation. Both EOs, at the highest concentrations tested, completely inhibited the germination and the radicle elongation of *Sinapis alba* L. *Lolium multiflorum* Lam. seed was less affected by the treatments, and the radicle elongation was more inhibited by the EOs than seed germination. Indeed, germination in some cases was stimulated, possibly indicating a hormesis phenomenon in the case of *Lactuca sativa* L. for EC and *Cucumis sativus* L.for EN when compared to the control. Among the crop plants, both EOs showed no significant inhibition of germination in all tested seeds, with the exception of *Raphanus sativus* L. The aromatic plant *Ruta graveolens* L. was the most sensitive, being significantly affected by the EOs both in germination and radicle elongation.

### 2.4. Antimicrobial Activity

The antimicrobial activity of EOs was tested on bacterial plant pathogens *Xanthomonas campestris* pv. *campestris (Xcc)*, *Enterobacter cloacae (E. cloacae)*, and *Citrobacter freundii* (*C. freundii*) of agricultural interest. At different time points (2, 6, 12, and 24 h) the minimum inhibitory concentration (MIC) was evaluated for the EOs (Table 5) compared to the principle pure components. Interestingly, all of the pure components tested showed a MIC value only when we used undiluted EOs [100% *v*/*v*], with data not shown.

The antibacterial activity depended on the concentration of the essential oil used (Table 5), whereby total inhibition was observed with undiluted EOs [100% *v*/*v*] for all the bacterial strains tested.

Moreover, differences were noted in the EO and the MIC target bacteria tested: the EO from EC showed a MIC of [0.01% *v*/*v*] on all three bacteria, whereas the EO from EN resulted in a MIC of [0.01% *v*/*v*] for *Xcc* and *E. cloacae* (Figure 4A,B), and a MIC of [0.1% *v*/*v*] for *C. freundii* (Figure 4C). The different antimicrobial activity against *C. freundii* could depend upon the exponential growth curve of this bacterial culture which was more rapid than that of the other two bacterial strains.

After 24 h of incubation, the essential oil from EC was the most efficient at all tested concentrations, whereas all pure components tested did not show relevant antimicrobial activity if tested in diluted doses (Figure 4A–C).

### 2.5. Antifungal Activity

The biological activity of the EOs and the pure compounds was also performed on the plant fungal pathogens *Fusarium oxysporum* f.sp. *lycopersici* and *Botrytis cinerea*. The results (Figure 5) indicated that the EO obtained from EC was less efficient in controlling the fungi than that from EN when used at the lowest concentration of 0.1% *v*/*v*. However, no significant differences between EC and EN purified principal compounds were noted at the higher concentration of 1% v/v, possibly due to the different concentrations of the compounds with the greater fungitoxic effect. This result was similar in the assay with the pure eucalyptol, which showed higher antifungal activity and thus greater inhibition could be attributed to the higher concentration of eucalyptol in the EO of EN compared to that of EC (79.5% and 67.7%, respectively). The pure α-terpineol and α-pinene demonstrated little efficacy in controlling the pathogenic fungi, although the effects were not constant, whereas the γ-terpinene appears to be a promising compound when compared to the two tested EOs. Among the fungal disease agents, *B. cinerea* showed a higher sensitivity to all the phytocompounds tested compared to *F. oxysporum* which was more resistant.

## 3. Discussion

The genus *Eucalyptus* has a high aesthetic and economic value for the floriculture sector and its fronds are increasingly used in the markets of Northern Europe for flower arrangements. Based on the data collected by the General Census of Agriculture, it was estimated that in 2005 eucalyptus production already accounted for 22% of the frond crops harvested in Western Liguria (provinces of Imperia and Savona) that went to the ornamental floral sector, with an extension of cultivation covering over 350 ha. The growing relevance of these species has since been reported by the flower market of Sanremo, whereby 50% of the green fronds marketed in the 2014–2015 period consisted of eucalyptus [10]. Thus, with the augmented eucalyptus production in recent years to meet the demand, there has also been a corresponding increase in the quantity of frond by-products derived from this activity. This fact raises the problem of the elimination of vegetative wastes, but, at the same time, this provides novel opportunities for the recovery and valorization of this plant material as a source of natural value-added products.

Micromorphological characterization is a fundamental step for the evaluation of the primary plant material that is crushed or powdered, which is necessary for the standardization or quality control of the plant by-product inputs to the recycling process. In these fragmented tissues, it is still possible to analyze the anatomical vegetative features, corroborate the conformity of the starting material, and note any adulterations or contamination from foreign organisms. In the genus *Eucalyptus*, the macromorphological characteristics of taxonomic value are the shape, color, and vein pattern of the leaves, while the important micromorphological features include the type of stomata, morphology of epicuticular waxes, presence of epidermal papillae more or less prominent, and the midrib section shape [11].

The phytochemical profile of the EO extracts obtained from the different species is important for identifying and quantifying the phytocompounds. The chemical composition of the essential oil in EC in the present study disagrees in part with the composition of essential oil of EC from Northwest Tunisia, in which high percentages of camphene (15.13%) and globulol (4.06%) were reported [12]. Instead, a greater similarity can be found with the EOs found in this species coming from Egypt [13], India [14], and Brazil [15]. It is possible to hypothesize that environmental conditions, and thus geographic locations, have a decisive influence on the composition of the EOs [16].

In our case, the composition of the EO found in EN agrees with the scarce findings in the scientific literature [17], thus demonstrating a novelty in the use of this species.

Furthermore, in both investigated species, the EOs indicated eucalyptol as the major component (67.7% in EC and 79.5% in EN). A similar composition regarding α-pinene and terpinene derivatives (α-terpinene, γ-terpinene, terpinolene, α-terpineol, and α-terpinyl acetate) as well as other main constituents were noted in EC and EN.

In general, the phytochemical profile of the EOs obtained from different plant species is important to compare their biological activities and to discriminate specific compounds with potential effects as antibacterial, antifungal, and herbicidal agents. For many plants essential oils, antibacterial and antifungal activity toward the pathogens affecting food or crops has been evidenced [18,19]. Whole essential oils, and many of their components, have been proposed as green pesticides [20,21]. The phytotoxicity of these essential oils was confirmed in the literature, whereby the EOs from different *Eucalyptus* species have been reported for their allelopathic and phytotoxic activities to various plants [22,23,24,25]. These biological activities can be primarily attributed to the high percentage of eucalyptol and terpinene derivatives for which phytotoxic properties have been reported [9,26,27,28]. In the earliest studies on allelopathic interactions in *Salvia leucuphylla* populations, eucalyptol was proposed as a volatile compound able to inhibit plant growth, thus influencing vegetation composition [29,30]. Some mechanisms of action were suggested [31], and the compound was proposed both for direct use as a bioherbicide and as a lead compound for herbicide synthesis [20,32].

In general, the EOs from the *Eucalyptus* species here investigated showed phytotoxicity to both weeds tested, which varied in relation to the concentration used and the species examined. On the contrary, both EOs showed no significant inhibition of seed germination of crop plants, except for *R. sativus*.

The EO yield from the leaf by-products of EC and the percent of its main component, eucalyptol, can be considered high in comparison with other eucalyptus species [33]. The EO of EC, based on its composition, showed the highest antimicrobial activity compared to EN. In particular, the presence in EC of δ-eIemene, α-terpinyl acetate, α-copaene, α-gurjunene, aromadendrene, α-humulene, and β-selinene compounds that are absent in the EO of EN, could contribute to increased antimicrobial activity against the Gram-negative bacteria tested. Moreover, the EO of EC has been reported to have antibacterial activity against many Gram-positive and Gram-negative bacteria and human pathogens [15,34], as well as against plant pathogenic bacteria [35]. Our results are similar to the findings in these previous studies and confirm the high antibacterial properties of the EO from EC.

The antifungal activity of terpenes can be directly related to the incorporation of these compounds into cellular membranes that compromises their structural integrity. On the other hand, differences in the lipid composition in the membrane of diverse fungi may be responsible for the action of these molecules [36]. Sterols are one class of lipids present in the membrane of fungi that may be targeted by these EOs [37]. For this reason, even if the EO of EN is composed only of 22 molecules, the highest concentration of eucalyptol (79.5%) may be sufficient to provide antifungal properties.

Several studies have highlighted the antimicrobial properties of the components analyzed in our study; in particular, the antimicrobial activity of α-pinene [38], terpinen-4-ol [39], and γ-terpinene [40] was determined and indicated by the different MIC values. Moreover, α-terpinyl acetate [41], present only in EC, has been classified as an antimicrobial compound, whereas, diversely, γ-terpinene did not show antimicrobial activity [42]. Interestingly, the present study showed that all pure components tested demonstrated an antimicrobial activity lower than that of EOs, suggesting a strong synergistic effect of the various constituents, which has been already reported in previous studies [43,44].

## 4. Materials and Methods

### 4.1. Chemicals

Pure standard components of EOs, eucalyptol, α-pinene, terpinen-4-ol, α-terpineol, α-tepinyl acetate, and γ-terpinene were purchased from Sigma Aldrich, Milano, Italy.

### 4.2. Plant Material

Small branches bearing juvenile foliage of *Eucalyptus cinerea* F. Muell. ex Benth (EC) and *Eucalyptus nicholii* Maiden and Blakely (EN) were collected in February 2022 at commercial plantations in the hinterland of Western Liguria (Finale Ligure, Savona, Italy) at Tovo San Giacomo (coordinates = 44.19683_ N, 8.26005_ E) and at Magliolo (coordinates = 44.191671_ N, 8.252987_ E), respectively (Figure 6). A voucher specimen of each species (GDOR n. 60106 and GDOR 60105, respectively) was deposited in the herbarium of the Department of Earth, Environment and Life Sciences, University of Genoa, Italy.

### 4.3. Macro- and Micromorphological Analyses

A macromorphological study of the leaves of each species was carried out with a LEICA M205 C stereomicroscope (Leica Microsystems, Wetzlar, Germany) to evaluate the shape, color, venation pattern, and presence of waxes and oil glands.

Leaf epidermal peelings were obtained from both the abaxial and adaxial surfaces using the nail polish technique [45]. For each species, five leaves were selected in sequence along a branch, starting from the first fully expanded juvenile leaf (i.e., from the third to the seventh internode). Afterward, six images of each selected leaf were captured at 10× magnification, and the oil gland density was estimated on an area corresponding to 0.962 mm^2^ using a Leica D.M. 2000 microscope equipped with a digital camera (DFC 320, Leica Microsystems, Wetzlar, Germany) [46]. All of the captured images were analyzed using the image processing software ImageJ [47], which enabled the measurements and counts required to obtain the quantitative data.

For the micromorphological analyses, transversal sections of fresh, healthy leaves were obtained by free-hand sectioning using a double-edged razor blade. Samples were mounted on glass slides in water and observed with a Leica DM2000 transmission light microscope, coupled with a computer-driven DFC320 digital camera (Leica Microsystems, Wetzlar, Germany).

For a more detailed analysis of the epidermal surface and the oil gland cavities within the mesophyll, small portions of the leaves were also observed using scanning electron microscopy (SEM). Leaves (about 1.5–2.0 cm^2^) were fixed in 70% ethanol–FineFix working solution (Milestone s.r.l., Bergamo, Italy) for 24 h at 4 °C, dehydrated through a series of increasing ethanol solutions [48], and critical-point dried in CO_2_ (CPD, K850 2M Strumenti s.r.l., Rome, Italy). Finally, dried samples were mounted on aluminum stubs using glued carbon tabs, sputter-coated with 10 nm gold [49], and observed with a Vega3 Tescan LMU SEM (Tescan USA Inc., Cranberry Twp, PA, USA) at an accelerating voltage of 20 kV.

### 4.4. Extraction of Essential Oils

The leaves of EC and EN were removed from the branches, cut into fragments, and then subjected to hydrodistillation using a Clevenger apparatus (Council of Europe, 2004), producing a pale yellow EOs extract. The EOs were dried on Na_2_SO_4_ and stored in sealed vials with nitrogen headspace in the dark, at 4 °C until analysis.

### 4.5. GC-FID Analyses

Analytical gas chromatography (GC) was carried out on a Perkin-Elmer Sigma-115 gas chromatograph (Perkin Elmer, Waltham, MA, USA) equipped with a flame ionization detector (FID) and a data handling processor. The separation was achieved using an HP5-MS fused-silica capillary column (30 m × 0.25 mm i.d., 0.25 μm film thickness, Agilent, Roma, Italy). The column temperature was 40 °C, with a 5 min initial hold, and then 270 °C at 2 °C/min, 270 °C (20 min), with the splitless injection mode (1 μL of a 1:1000 n-hexane solution). Injector and detector temperatures were 250 °C and 290 °C, respectively. Analysis was also run by using a fused silica HP Innowax polyethylene glycol capillary column (50 m × 0.20 mm i.d., 0.25 μm film thickness, Agilent, Roma, Italy). In both cases, helium was used as the carrier gas (1.0 mL/min).

### 4.6. GC/MS Analyses

Analysis was performed on an Agilent 6850 Ser. II apparatus (Agilent, Roma, Italy), fitted with a fused silica DB-5 capillary column (30 m × 0.25 mm i.d., 0.33 μm film thickness, Agilent, Roma, Italy), coupled to an Agilent Mass Selective Detector MSD 5973: ionization energy voltage 70 eV and electron multiplier voltage energy 2000 V. Mass spectra (MS) were scanned in the range 40–500 amu, scan time 5 scans/s. Gas chromatographic conditions were as reported in the previous paragraph with a transfer line temperature of 295 °C.

### 4.7. Identification of the Essential Oil Components

Most constituents were identified by GC by comparison of their Kovats retention indices (Ri) [determined relative to the retention times (tR) of n-alkanes (C_10_–C_35_)], comparing values with either those of the literature [50,51,52,53] and mass spectra on both columns or those of authentic compounds available in our laboratories by means of NIST 02 and Wiley 275 libraries [54]. The components’ relative concentrations were obtained through peak area normalization. No response factors were calculated.

### 4.8. Phytotoxic Activity

The phytotoxic activity was evaluated on seed germination and radicle emergence/elongation of several plant species: weeds (*Lolium multiflorum* Lam., *Sinapis alba* L.), horticultural crops (*Raphanus sativus* L., *Pisum sativum* L., *Cucumis sativus* L., *Lactuca sativa* L.), and the aromatic plant *Ruta graveolens* L. These seeds were selected for their easy and well-known germinability. *R. sativus*, *L. sativa*, *P. sativum*, *R. graeolens*, and *C. sativus* seeds were purchased from Blumen group SRL (Emilia-Romagna, Italy); *L. multiflorum* seeds were purchased from Fratelli Ingegnoli Spa (Milano, Italy); and the seeds of *S. alba* were collected from a wild population near the university campus in Fisciano (Salerno, Italy). The seeds were surface sterilized in 95% ethanol for 15 s and sown in Petri dishes (Ø = 90 mm) on three layers of Whatman filter paper. They were impregnated with 7 mL of deionized water used as a first control to verify the germinability of the seeds, then 7 mL of a water–acetone mixture (99.5:0.5, *v*/*v*) was used as a second control since EOs were dissolved in this mixture due to their lipophilicity, or 7 mL of the tested solution at different concentration doses (1000, 500, 250 and 100 μg/mL). Controls carried out with a water–acetone mixture alone showed no differences in comparison to controls in water alone. The germination conditions were 20 ± 1 °C, with a natural photoperiod. Seed germination was checked in Petri dishes every 24 h. A seed was considered germinated when the protrusion of the root became evident [55]. On the fifth day (after 120 h) for *R. sativus* and on the tenth day (after 240 h) for the other tested seeds, the effects on radicle elongation were determined by measuring the root length in cm. Each evaluation was replicated three times, using Petri dishes containing 10 seeds each. Data were expressed as the mean ± standard deviation for both germination and radicle elongation.

### 4.9. Plant Pathogens

The EOs were tested against three Gram-negative bacterial pathogens: *Xanthomonas campestris* pv. *campestris*, the causal agent of black rot disease that causes important crop losses in the Brassicaceae, and *Enterobacter cloacae* and *Citrobacter freundii*, which are clinical pathogens but often found as emerging plant pathogens. In addition, biocontrol activity was also tested on two fungal phytopathogens: the soilborne vascular fungus *Fusarium oxysporum* f. sp. *lycopersici* (4287) (*F. oxysporum*) and the polyphagous foliar and soft rot pathogen *Botrytis cinerea* (*B. cinerea*).

### 4.10. Antimicrobial Activity

Dilution broth susceptibility assays were used for the antimicrobial evaluation of the bacteria [56]. Briefly, stock solutions of the EOs and standards were prepared in ethanol by mixing 1 mL of the EOs or the compound standard with 9 mL of EtOH in test tubes to obtain the mother solution, then serial dilutions were prepared (100%, 10%, 0.1%, and 0.01% *v*/*v*) in distilled water. The negative control was prepared by mixing 1 mL of distilled water with 9 mL of alcohol solution. An amount of 1 mL of each dilutant and 0.5 mL of the bacterial cultures in an exponential growth phase (10^8^ CFU/mL) were added to 8 mL of Nutrient Broth (Sigma Aldrich, Milano, Italy) with a final volume of 9.5 mL, and incubated under shaking conditions at 37 °C for 2, 6, 12, and 24 h. After each incubation, bacterial concentrations were measured via a Model 680 Microplate Reader at 600 nm (Bio-rad laboratories, Segrate, Italy) [57].

### 4.11. Antifungal Activity

The EOs and commercial standards were diluted ten times with ethanol (*v*/*v*), followed by serial dilutions in distilled sterile water (1% and 0.1% *v*/*v*). Negative controls were prepared similarly by substituting EOs with distilled sterile water. To measure the growth inhibitory activity of EOs on *F. oxysporum* and *B. cinerea*, 190 µL of freshly obtained conidia were resuspended in GM medium [58], adjusted to pH 5, containing a final concentration of 3.19 × 10^6^ conidia mL^−1^ of the fungus, and then supplemented with 10 µL of each diluted EO. All samples were cultured in 96-well plates (Falcon^®^ 96-well Clear Flat Bottom TC-treated Culture Microplate) for 16 h at 28 °C under shaking conditions. Optical density at 600 nm (O.D. 600) was measured at 10 min intervals using a Tecan Infinite 200 Pro microplate reader (Tecan, Männedorf, Switzerland).

### 4.12. Statistical Analysis

Results were analyzed by one-way analysis of variance (ANOVA) using GraphPad Prism 6.0 (Software Inc., San Diego, CA, USA), expressed as means ± standard deviation (S.D.) of three independent experiments (*n* = 3 for micromorphological and phytotoxicity assays) and analyzed by one-way analysis of variance (ANOVA) using GraphPad Prism 6.0 (Software Inc., San Diego, CA, USA). Results were considered significant at *p* < 0.05.

## 5. Conclusions

To date, most investigations on the antimicrobial effects of essential oils have been focused on human pathogens, with little interest demonstrated in the potential control of phytopathogenic microorganisms. However, the essential oils from both *Eucalyptus* species tested in the present study, containing a high percentage of eucalyptol and of terpinene derivatives, showed—to different extents—selective phytotoxicity, as well as antibacterial and antifungal activities against bacteria and fungi causing plant diseases.

These data indicate that the bioactive metabolites found in the essential oils from *E. cinerea* and *E. nicholii* by-products could be directly extracted and provide opportunities for the reuse of the pruning biomass to obtain new agrochemicals, thus transforming the waste from a cost loss to a value-added resource.

On the other hand, it is also well known that essential oils can exert biological effects on various plant species and soil microorganisms, with potential risks to biodiversity. Therefore, it is highly desirable to implement careful management of these biomasses, avoiding their indiscriminate reintroduction into the environment, and to have an in-depth study of the biological effects of their active metabolites.

Our innovative approach meets the principles of the circular economy and current directives aimed at developing a suitable method to discard agricultural by-products, reducing the negative impacts of chemical pesticides both on the agrosystems and on public health.

## Figures and Tables

**Figure 1 plants-11-02777-f001:**
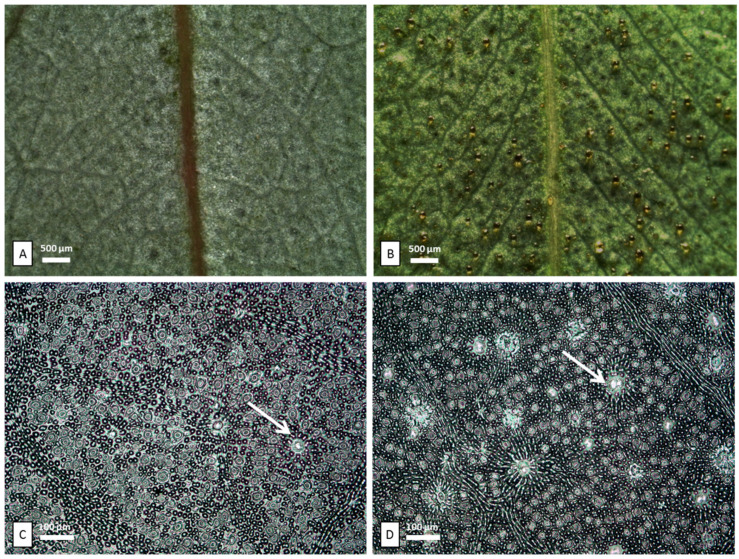
Stereomicroscopy: abaxial surfaces of *E. cinerea* (**A**) and of *E. nicholii* (**B**). Light microscopy: leaf epidermal peelings obtained via the nail polish technique from abaxial surfaces of *E. cinerea* (**C**) and of *E. nicholii* (**D**). White arrows show the overlying cells covering the secretory cavities.

**Figure 2 plants-11-02777-f002:**
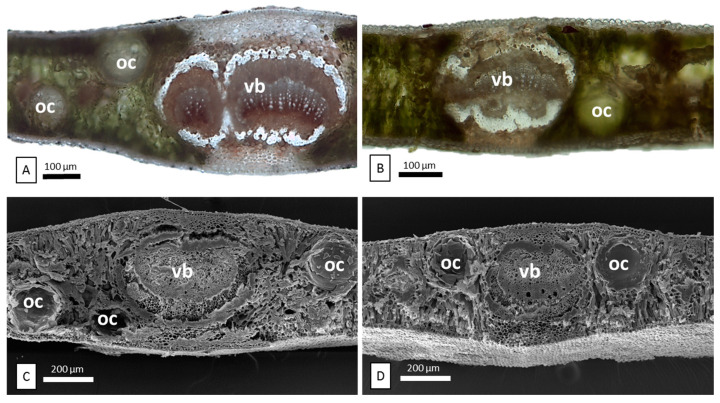
Light microscopy and SEM images.Transverse sections through leaf midrib of *E. cinerea* (**A**,**C**) and *E. nicholii* (**B**,**D**) showing vascular bundles (vb) and oil cavities (OC) within the mesophyll.

**Figure 3 plants-11-02777-f003:**
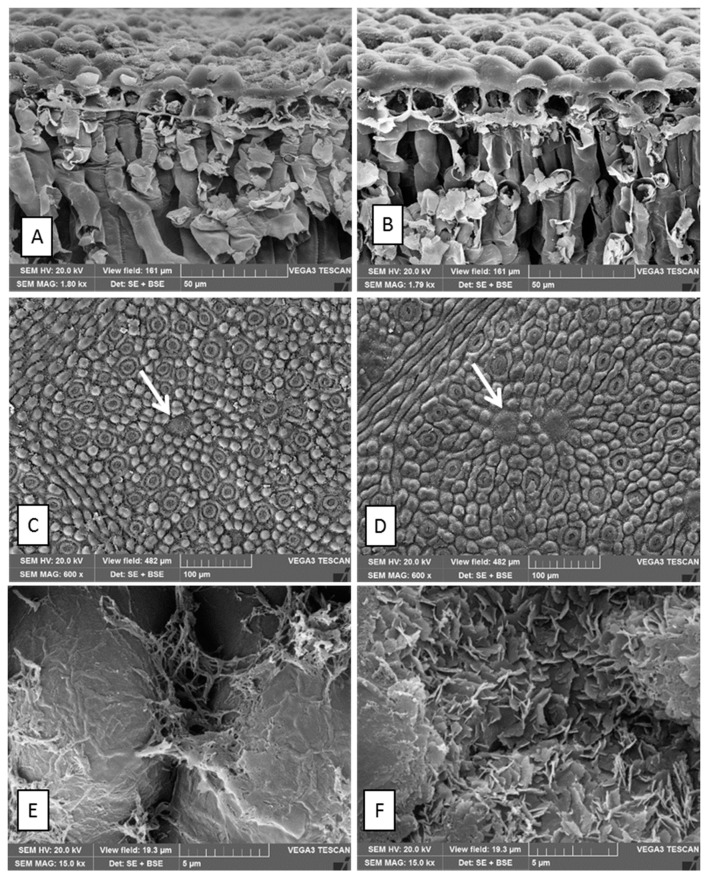
SEM images (**A**,**C**,**E**—*E. cinerea* and **B**,**D**,**F**—*E. nicholii*): (**A**,**B**) leaf transversal section showing papillae on the adaxial surface and (**C**,**D**) adaxial surfaces. White arrows show the overlying cells covering the secretory cavities. (**E**,**F**) tubular waxes in EC and crystalloid waxes (rosettes) in EN.

**Figure 4 plants-11-02777-f004:**
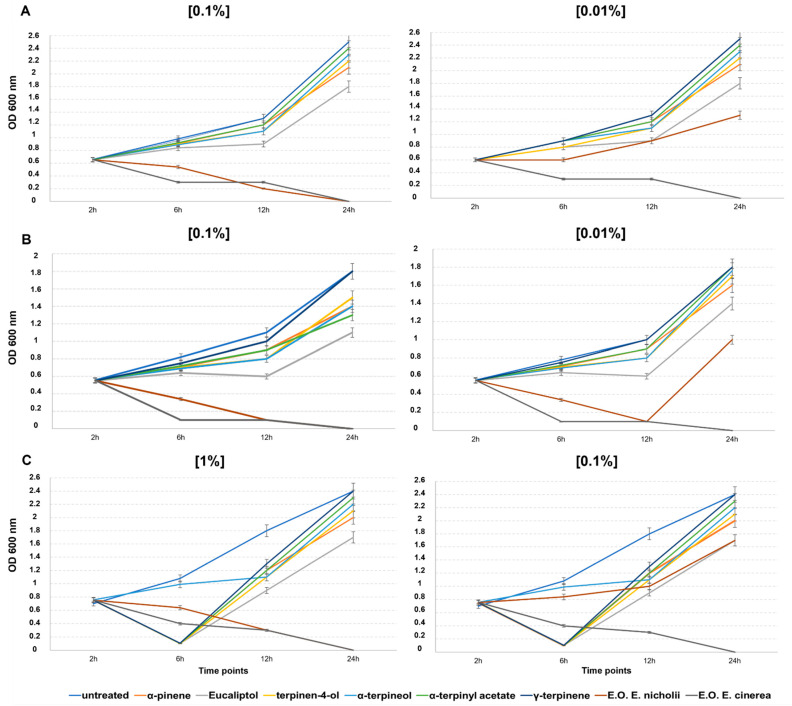
Antibacterial activity of different concentrations (0, 0.01, 0.1, and 1% *v*/*v*) of EOs extracted from *E. cinerea* or *E. nicholii*, and six pure EO compounds on the culture growth of: (**A**) *Xanthomonas campestris* pv. *campestris*, (**B**) *Enterobacter cloacae*, and (**C**) *Citrobacter freundii* 24 h after treatment.

**Figure 5 plants-11-02777-f005:**
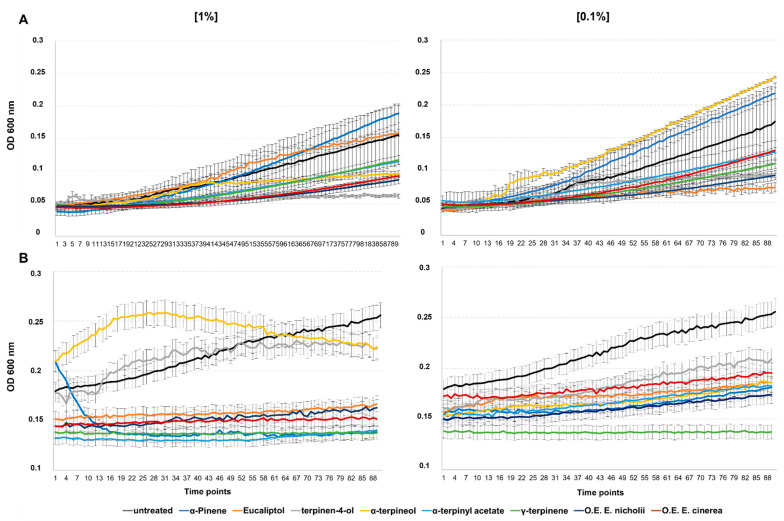
Antifungal activity of EOs from *E. cinerea* and *E. nicholii* and six pure compounds applied at different concentrations to the fungal pathogens: (**A**) *Fusarium oxysporum* and (**B**) *Botritis cinerea*. Measurements were conducted at 10 min and started after 80 min of incubation.

**Figure 6 plants-11-02777-f006:**
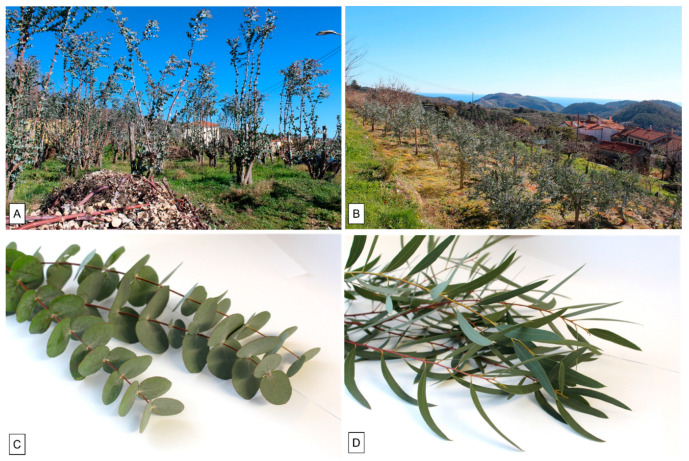
Plantations and collected branches of *E. cinerea* (**A**,**C**) and of *E. nicholii* (**B**,**D**).

**Table 1 plants-11-02777-t001:** *E. cinerea* and *E. nicholii* oil gland mean density on the leaf surfaces (equivalent to the number of glands per mm^2^) and yield (%) of EOs obtained from leaves by hydrodistillation.

Species	Oil Gland Mean Density (No. Glands per mm^2^ ± DS)		%Yield of Essential Oil
	Abaxial	Adaxial	
*E. cinerea*	9.6 ± 3.2	12.2 ± 2.4	2.56
*E. nicholii*	22.1 ± 4.9	18.3 ± 4.2	3.67

Statistically significant differences: interspecies EC abaxial vs. EN abaxial *p* < 0.0001, and EC adaxial vs. EN adaxial *p* < 0.0001; intraspecies EC abaxial vs. adaxial *p* < 0.001 and EN abaxial vs. adaxial *p* < 0.01.

**Table 2 plants-11-02777-t002:** Percent composition of the essential oils extracted from the leaves of *E. cinerea* (EC) and *E. nicholii* (EN) by-products as determined via gas chromatography–mass spectra analysis.

Compound	EC %	EN %	Ki ^a^	Ki ^b^	Identification ^c^
α-Pinene	7.3	3.7	858	1028	1,2,3
Camphene	0.2	0.2	869	1075	1,2,3
β-Pinene	0.3	0.2	893	1105	1,2,3
3-*p*-Menthene	0.1	0.2	903		1,2
β-Myrcene	0.2	0.1	913	1174	1,2,3
α-Phellandrene	0.8	0.8	921	1176	1,2,3
α-Terpinene	3.3	3.3	934	1188	1,2,3
Eucalyptol (1,8-cineole)	67.7	79.5	949	1213	1,2,3
γ-Terpinene	1.2	0.9	971	1254	1,2,3
*p*-Mentha-3,8-diene	0.2	0.1	982	1259	1,2
Terpinolene	3.5	2.5	998	1265	1,2,3
Linalool	0.2	0.3	1018	1553	1,2,3
*allo-*Ocimene	0.1	-	1039	1409	1,2
*trans*-Pinocarveol	-	0.3	1041	1641	1,2
*neo-allo*-Ocimene	0.3	-	1042		1,2
Pinocarvone	-	0.1	1063	1587	1,2
Borneol	0.1	0.1	1067	1715	1,2,3
*neoiso*-Pulegol	0.2	0.1	1070		1,2
Terpinen-4-ol	0.6	0.7	1079	1705	1,2,3
α-Terpineol	3.9	3.9	1093	1720	1,2,3
Linalyl acetate	0.1	-	1219	1565	1,2
δ-EIemene	-	0.1	1220	1480	1,2,3
α-Terpinyl acetate	5.2	-	1234	1687	1,2
α-Copaene	0.3	-	1270	1498	1,2,3
α-Gurjunene	0.1	-	1287	1529	1,2
(E)-Caryophyllene	0.5	0.3	1295	1575	1,2,3
Aromadendrene	0.3	-	1308	1628	1,2
α-Humulene	0.1	-	1323	1651	1,2
*allo*-Aromadendrene	0.1	0.2	1330	1661	1,2
β-Selinene	0.2	-	1347	1697	1,2
Viridiflorene	0.5	0.1	1366	1713	1,2
**Total**	**97.6**	**97.7**			
Monoterpene hydrocarbons	17.5	12.0			
Oxigenated monoterpenes	78.0	85.0			
Sesquiterpene hydrocarbons	2.1	0.7			
Oxigenated sesquiterpenes	0	0			

^a,b^ The Kovats retention indices determined relative to a series of n-alkanes (C10-C35) on the apolar HP5-MS and the polar HP Innowax column, respectively. ^c^ identification method: 1 = comparison of the Kovats retention indices with published data, 2 = comparison of mass spectra with those listed in the NIST 02 and Wiley 275 libraries and with published data, and 3 = co-injection with authentic compounds.

**Table 3 plants-11-02777-t003:** Phytotoxic activity of the *E. cinerea* essential oil applied at different concentrations (µg/mL) to seeds of diverse plant species.

Number of Germinated Seeds							
	*L. multiflorum*	*S. alba*	*C. sativus*	*L. sativa*	*P. sativum*	*R. sativus*	*R. graveolens*
Control	6.3 ± 1.5	8.3 ± 1.5	9.3 ± 1.2	7.3 ± 1.5	9.7 ± 0.6	6.3 ± 1.2	6.3 ± 2.5
Treatment (µg/mL)							
100	3.3 ± 2.1	6.0 ± 1.7	8.7 ± 0.6	8.0 ± 2.0	8.7 ± 2.3	2.0 ± 1.0 **	1.0 ± 1.0 **
250	8.0 ± 1.0	1.0 ± 1.0 ****	9.7 ± 0.6	9.0 ± 1.0	8.7 ± 1.5	2.0 ± 2.0 **	0.3 ± 0.6 ***
500	6.0 ± 3.6	0.7 ± 1.2 ****	10.0 ± 0.0	7.3 ± 1.5	8.7 ± 1.5	0.0 ± 0.0 ***	2.0 ± 1.0 **
1000	6.7 ± 1.5	0.0 ± 0.0 ****	8.7 ± 1.2	8.0 ± 0.0	8.0 ± 1.0	0.3 ± 0.6 ***	0.3 ± 0.6 ***
**Radicle Length (cm)**							
	*L. multiflorum*	*S. alba*	*C. sativus*	*L. sativa*	*P. sativum*	*R. sativus*	*R. graveolens*
Control	1.2 ± 0.6	0.4 ± 0.2	6.3 ± 0.9	3.2 ± 1.2	6.0 ± 1.8	3.1 ± 1.0	1.7 ± 0.8
Treatment (µg/mL)							
100	0.4 ± 0.0	0.2 ± 0.1	3.4 ± 0.7 ***	1.4 ± 0.5	3.5 ± 1.3	2.2 ± 0.0 **	0.0 ± 0.0 **
250	0.9 ± 0.4	0.0 ± 0.0 **	2.3 ± 0.5 ****	2.5 ± 0.7	2.4 ± 0.8 *	0.0 ± 0.0 **	0.0 ± 0.0 **
500	1.1 ± 0.5	0.0 ± 0.0 **	2.3 ± 0.4 ****	1.8 ± 0.7	2.9 ± 0.9 *	0.0 ± 0.0 **	0.9 ± 0.0
1000	0.8 ± 0.4	0.0 ± 0.0 **	2.3 ± 0.8 ****	2.3 ± 0.8	2.8 ± 1.0 *	0.0 ± 0.0 **	0.0 ± 0.0 **

Significant differences among the dose concentrations of the EO treatments: * *p* < 0.05; ** *p* < 0.01; *** *p* < 0.001; **** *p* < 0.00001.

**Table 4 plants-11-02777-t004:** Phytotoxic activity of the *E. nicholii* essential oil applied at different concentrations (µg/mL) to seeds of diverse plant species.

Number of Germinated Seeds							
	*L. multiflorum*	*S. alba*	*C. sativus*	*L. sativa*	*P. sativum*	*R. sativus*	*R. graveolens*
Control	5.3 ± 2.1	9.7 ± 0.6	9.0 ± 1.0	8.7 ± 0.6	10.0 ± 0.0	6.0 ± 1.0	8.0 ± 2.0
Treatment (µg/mL)							
100	5.7 ± 1.5	6.0 ± 1.0 ***	10.0 ± 0.0	8.0 ± 0.0	9.3 ± 0.6	2.0 ± 1.0 **	4.7 ± 3.5
250	4.3 ± 1.2	4.0 ± 1.0 ****	10.0 ± 0.0	8.7 ± 0.6	9.0 ± 0.0	2.0 ± 1.0 **	1.0 ± 1.0 **
500	5.3 ± 3.2	0.0 ± 0.0 ****	9.0 ± 1.0	9.0 ± 1.7	9.7 ± 0.6	0.7 ± 0.6 ***	1.0 ± 1.0 **
1000	5.7 ± 2.3	0.0 ± 0.0 ****	10.0 ± 0.0	8.0 ± 1.0	9.0 ± 1.0	0.7 ± 1.2 ***	0.0 ± 0.0 **
**Radicle Length (cm)**							
	*L. multiflorum*	*S. alba*	*C. sativus*	*L. sativa*	*P. sativum*	*R. sativus*	*R. graveolens*
Control	3.4 ± 0.8	0.6 ± 0.2	6.7 ± 0.9	2.7 ± 1.0	6.3 ± 1.9	2.8 ± 1.1	1.2 ± 0.5
Treatment (µg/mL)							
100	0.9 ± 0.3 ***	0.4 ± 0,2	2.6 ± 0.7 ***	2.6 ± 0.8	3.4 ± 0.9 *	1.3 ± 0.0 **	0.8 ± 0.0
250	0.6 ± 0.2 ****	0.2 ± 0.1 *	2.7 ± 0.5 ***	1.5 ± 0.5	2.9 ± 0.7 *	1.9 ± 0.0	0.0 ± 0.0 ***
500	0.6 ± 0.2 ****	0.0 ± 0.0 ***	1.9 ± 0.6 ****	2.6 ± 0.9	3.7 ± 1.2	0.0 ± 0.0 ****	0.0 ± 0.0 ***
1000	0.9 ± 0.4 ***	0.0 ± 0.0 ***	2.0 ± 0.9 ****	2.4 ± 0.8	2.9 ± 1.2 *	0.0 ± 0.0 ****	0.0 ± 0.0 ***

Significant differences among the dose concentrations of the EO treatments: * *p* < 0.05; ** *p* < 0.01; *** *p* < 0.001; **** *p* < 0.00001.

**Table 5 plants-11-02777-t005:** Minimum inhibitory concentration (MIC) of the EOs extracted from *E. cinerea* and *E. nicholii* tested at different concentrations [% *v*/*v*].

Treatment	MIC (% *v*/*v*)		
	*Xanthomonas campestris* pv. *campestris*	*Enterobacter* *cloacae*	*Citrobacter* *freundii*
EO of *E. cinerea* (EC)	0.01%	0.01%	0.01%
EO of *E. nicholii* (EN)	0.01%	0.01%	0.1%

## Data Availability

Not applicable.

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
