# Peer review of "Eucalyptus cinerea and E. nicholii by-Products as Source of Bioactive Compounds for Agricultural Applications"

_plants, 2022, doi:10.3390/plants11202777_

Round 1

Reviewer 1 Report

The manuscript plants-1973578 entitled " Eucalyptus cinerea and E. nicholii by-products as a source of bioactive compounds for agricultural applications" reported the chemical composition, the antibacterial and antifungal activities of essential oils obtained from Eucalyptus cinerea and Eucalyptus nicholii by-products cultivate in Liguria (Italy).

This is great research work. It is well written; the introduction section does explain what was done on the topic analyzed. In the same way, a deep review of pertinent literature was used. The materials and methods are very clear allowing their reproduction in any laboratory. The scientific quality of the results and discussion section is so good. The results are presented in a clear and concise way. The authors discuss and compare their findings with the previous report on this field.

The only serious issue to improve will be The English language. In its current state, the level of English throughout your manuscript does not meet the journal's desired standard. There are some grammatical and spelling errors and full stops missing as well as instances of badly worded/constructed sentences. Please check the manuscript and refine the language carefully. I suggest that you should ask several colleagues who are skilled authors to check the English before your submission.

Additionally, the main content of the abstract should include the brief purpose of the research, the principal result, and the major conclusion. The abstract, in the present form, is not adequate. Additionally, (i) the authors must state the revised justification in the abstract to support the study; (ii) some results should be nice.

Author Response

  • The manuscript plants-1973578 entitled " Eucalyptus cinerea and E. nicholii by-products as a source of bioactive compounds for agricultural applications" reported the chemical composition, the antibacterial and antifungal activities of essential oils obtained from Eucalyptus cinerea and Eucalyptus nicholii by-products cultivate in Liguria (Italy).

 This is great research work. It is well written; the introduction section does explain what was done on the topic analyzed. In the same way, a deep review of pertinent literature was used. The materials and methods are very clear allowing their reproduction in any laboratory. The scientific quality of the results and discussion section is so good. The results are presented in a clear and concise way. The authors discuss and compare their findings with the previous report on this field.

The only serious issue to improve will be The English language. In its current state, the level of English throughout your manuscript does not meet the journal's desired standard. There are some grammatical and spelling errors and full stops missing as well as instances of badly worded/constructed sentences. Please check the manuscript and refine the language carefully. I suggest that you should ask several colleagues who are skilled authors to check the English before your submission.

We thank the reviewer for his/her comment. We checked all the manuscript and refined the language.

Additionally, the main content of the abstract should include the brief purpose of the research, the principal result, and the major conclusion. The abstract, in the present form, is not adequate. Additionally, (i) the authors must state the revised justification in the abstract to support the study; (ii) some results should be nice.

We thank the reviewer for his/her comment.  As the reviewer suggested we rewrite totally the abstract according to the author's guidelines and the journal.

Reviewer 2 Report

The paper entitled “Eucalyptus cinerea and E. nicholii by-products as source of bio-active compounds for agricultural applications” reports an interesting work concerning the phytochemical and bioactivities evaluation of the essential oils obtained from two different pruning wastes.

The manuscript is well written, clear in language and scientifically solid. The overall impact of the work can be limited by the specificity of the considered cultivars, till it keeps a good scientific value for the field esperts.

I suggest just a check of the format in some points such as the use of bold style in tables 3 and 4.

For these reasons i consider the submitted paper suitable for pubblication.

Author Response

  • The paper entitled “Eucalyptus cinerea and E. nicholii by-products as source of bio-active compounds for agricultural applications” reports an interesting work concerning the phytochemical and bioactivities evaluation of the essential oils obtained from two different pruning wastes.

The manuscript is well written, clear in language and scientifically solid. The overall impact of the work can be limited by the specificity of the considered cultivars, till it keeps a good scientific value for the field esperts.

I suggest just a check of the format in some points such as the use of bold style in tables 3 and 4.

We thank the reviewer for his/her comment. We checked all the tables in the manuscript and change the style as the journal suggests.

For these reasons i consider the submitted paper suitable for pubblication.

Reviewer 3 Report

The paper is well structured, the bibliography used covers the topic and the key message is well depicted. The experimental results are carefully demonstrated.

Some, minor revisions are necessary.

1 What do the values in Fig 3 represent ? number of germinated seeds and radicle length in cm?

2.  Legend in Figure 4  is not clear. Use larger size font.

3.  Discussion should be more comprehensive.

4. Conclusion is too extended, it should be more concise

To sum up, I think the paper can be accepted after minor revision

Author Response

  • The paper is well structured, the bibliography used covers the topic and the key message is well depicted. The experimental results are carefully demonstrated.

Some, minor revisions are necessary.

  • What do the values in Fig 3 represent? number of germinated seeds and radicle length in cm?

We thank the reviewer for his/her comment. Probably is not the figure but table 3 that we have modified as suggested.

  • Legend in Figure 4 is not clear. Use larger size font.

 As the reviewer suggested we modified the font legend in Figures 4 and 5.

  • Discussion should be more comprehensive.

As recommended, we have modified the discussion to make it more comprehensive.

  • Conclusion is too extended, it should be more concise

We have rewritten the conclusion and made it more concise.

  • To sum up, I think the paper can be accepted after minor revision